# Independent Aridity and Drought Pieces of Evidence Based on Meteorological Data and Tree Ring Data in Southeast Banat, Vojvodina, Serbia

**Milivoj B. Gavrilov [1,\*], Wenling An [2,3], Chenxi Xu [2,3], Milica G. Radaković [1], Qingzhen Hao [2,3,4], Fan Yang [2,4], Zhengtang Guo [2,3,4], Zoran Perić [5], Gavrilo Gavrilov [1] and Slobodan B. Marković [1,6]**

1   Chair for Physical Geography, Faculty of Science, University of Novi Sad, Trg Dositeja Obradovića 3, 21000 Novi Sad, Serbia; milica.radakovic.95@gmail.com (M.G.R.); gavrilogavrilov365@gmail.com (G.G.); baca.markovic@gmail.com (S.B.M.)

2   Key Laboratory of Cenozoic Geology and Environment, Institute of Geology and Geophysics, Chinese Academy of Sciences, Beijing 100029, China; wlan@mail.iggcas.ac.cn (W.A.); cxxu@mail.iggcas.ac.cn (C.X.); haoqz@mail.iggcas.ac.cn (Q.H.); fyang@mail.iggcas.ac.cn (F.Y.); ztguo@mail.iggcas.ac.cn (Z.G.)

3   CAS Center for Excellence in Life and Paleoenvironment, Beijing 100044, China

4   University of Chinese Academy of Sciences, Beijing 100049, China

5   Research Group for Terrestrial Paleoclimates, Max Planck Institute for Chemistry, Hahn Meitner Weg 1, 55128 Mainz, Germany; zoranperic.geo@gmail.com

6   Serbian Academy of Sciences and Arts, Knez Mihajlova 35, 11000 Belgrade, Serbia

\*   Correspondence: gavrilov.milivoj@gmail.com

**Abstract:** In this study, aridity data and tree ring data were collected in Northern Serbia, in Southeast (SE) Banat, a subregion within Vojvodina, and Vojvodina at large. They were each investigated independently. The De Martonne Aridity Index and the Forestry Aridity Index are derived from examining the relationship between precipitation and surface air temperature data sets sourced from seven meteorological stations in SE Banat, and from 10 meteorological stations located in Vojvodina as a whole. Vojvodina is a large territory and used as the control area, for the period 1949–2017. The Palmer Drought Severity Index was derived for the period 1927–2016, for both SE Banat and the totality of Vojvodina. The results of the Tree Ring Width Index were obtained from samples collected in or around the villages of Vlajkovac and Šušara, both located in SE Banat, for the period 1927–2017. These tree ring records were compared with three previous aridity and drought indices, and the meteorological data on the surface air temperature and the precipitation, with the objective being to evaluate the response of tree growth to climate dynamics in the SE Banat subregion. It was noted that the significant positive temperature trends recorded in both areas were too insufficient to trigger any trends in aridity or the Tree Ring Width Index, as neither displayed any change. Instead, it appears that these climatic parameters only changed in response to the precipitation trend, which remained unchanged during the investigated period, rather than in response to the temperature trend. It appears that the forest vegetation in the investigated areas was not affected significantly by climate change in response to the dominant temperature increase.

**Keywords:** De Martonne aridity index; forestry aridity index; Palmer drought severity index; tree ring width index; temperature; precipitation; trend; correlation

## 1. Introduction

In recent decades, the issues of climate variability and/or climate change have been the focus of many scientific studies. Global climate change, caused by natural processes as well as anthropogenic factors, is a major environmental issue that may have a significant impact on the world over the course of the 21st century [1–4]. Governments, the scientific community, the media, and people all over the world have been paying more and more attention to recent trends in global climate change [5]. Temperature change [6], precipitation change [7] and the rate of these changes are some of the most important drivers of climate change. It seems to be more useful to analyze temperature and precipitation simultaneously, which is most similar to their impact on climate. Parameters in which there is a mathematical quotient/ratio of precipitation (or values of air humidity) and temperature are known as aridity indices and are used as a measure of aridity. According to the American Meteorological Society [8], aridity is the degree to which a climate lacks effective, life-promoting moisture, or, in the climate sense of the term, aridity is the opposite of humidity. Furthermore, the higher the values of the aridity indices in any given region, the greater the water resource variability [9]. Therefore, aridity indices are not only important as an indicator of plant growth but could also be regarded as good indicators of climate changes, and can be used to estimate changes in runoff [10].

The most renowned aridity parameter is the De Martonne [11], (DM), Aridity Index. This index can be calculated for different time scales, such as months, seasons and years. The DM index is used worldwide in order to identify the dry/humid climate conditions of any given region or regions [9,12–16], and provides a satisfactory description of the climate regions and climatic changes in the territory of Serbia, for the period 1949–2015 as well [17–19]. Similar to the DM index, the Pinna [20] combinative aridity index can be calculated using the annual amount of precipitation and the mean annual surface temperature. However, unlike the DM index, it only provides the annual aridity. This index is also quite widely used and can be found in the previously cited references. However, some scientists prefer to identify aridity conditions employing indices based on (potential or reference) evapotranspiration, calculated using different formulas [21–26]. For instance, for the classification of continental and oceanic climates, the Johansson [27] continentality index and the Kerner [28] oceanity index are used [9,12,15].

There is also a class of aridity indices that are based on relevant data associated with vegetation processes. One of the oldest of these is the Emberger [29] index, which is obtained by tracking the mean annual precipitation as well as the mean temperature of both the coldest and hottest months. The FAO (Food and Agriculture Organization) [30] aridity index is the ratio between the total annual precipitation and potential evapotranspiration. One of the most recent indices of the vegetative class is the Forestry Aridity Index (*FAI*), published by Führer [31]. The *FAI* is specifically designed to describe aridity conditions in Hungary, and was later successfully applied to the Vojvodina region in Northern Serbia [32]. The Palmer Drought Severity Index (PDSI) [22] was also utilized in this study. The PDSI has been widely used to evaluate the hydro climatic status of many areas [33]. In addition, the tree ring width has been used in evaluating regional aridity in Serbia and surrounding areas [34,35]. In the present study, we also used a Tree Ring Width Index (TRWI) series in the investigated regions. We examine the relationship between tree ring width and local climatic parameters, in order to evaluate the efficacy of the TRWI in evaluating the aridity.

In this study, the focus will be on two types of approach. In the first approach, aridity will be explored through the following values: the DM Aridity Index, and the *FAI*, all obtained from meteorological data from the investigated regions of the larger Vojvodina, and of Southeastern (SE) Banat, for the period 1949–2017, and PDSI for both of the investigated regions for the period 1927–2016. In the second approach, the TRWI value will be explored using sample data from SE Banat for the period 1927–2017. The differing test periods for the different applied indexes are determined by the maximum length of time series data for each reference index. The results of these two groups of investigations will be compared with the surface air temperature and precipitation data. Overall, the obtained results can be considered representative, as they can be an indicator for recent climate changes in both Southeast Banat and Vojvodina, as well as for the wider, relatively uniform geomorphologic area located in

central Europe known as the Carpathian (Pannonian) Basin (CB). Correspondingly, these results may prove to be particularly useful in evaluating the impact of climate change on forest vegetation, given that, in the analyzed period of 1949–2017, there exist more than two, 30-year climatic cycles. Finally, the characteristics of recent aridity/drought are interpreted in the context of paleoclimatic and paleoenvironmental phenomena, during the Late Pleistocene, in the investigated region.

## 2. Study Area and Data

### 2.1. Study Area

In this study, SE Banat (1153 km$^2$) is the primary research area. This subregion is represented with two municipalities: Bela Crkva (353 km$^2$) and Vršac (800 km$^2$). The research areais situated in the southeastern part of Vojvodina (21,506 km$^2$), and encompass the Danube, Sava, Tisa, Karaš, and Nera rivers (Figure 1).

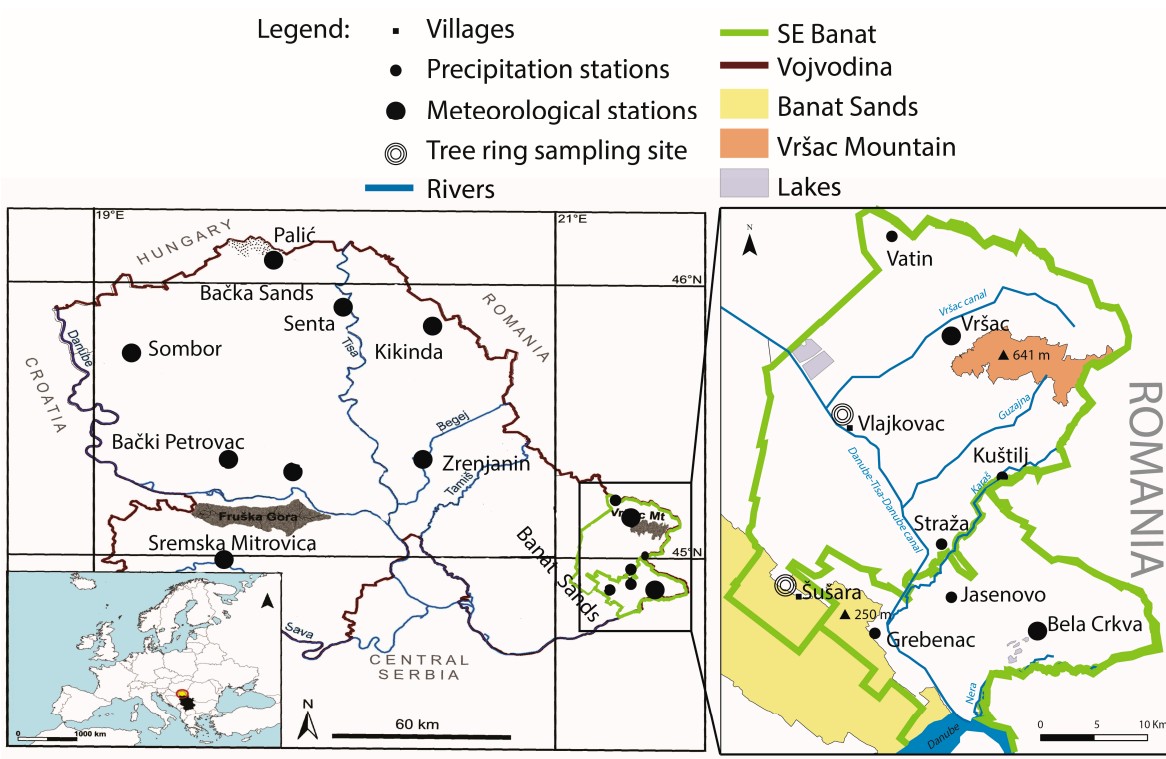

**Figure 1.** Territories of Vojvodina and SE Banat, their locations in Serbia and Europe, meteorological stations and tree ring sampling site used in this study.

Vojvodina is a lowland region where more than 60% of the area is covered by loess and loess-like sediments [36]. The loess–palaeosol sequences situated there appear to be the most detailed archive of Middle and Late Pleistocene climatic and environmental fluctuations on the European continent [37–39]. The most distinctive landforms in the Vojvodina region are two mountains: Fruška Gora Mountain (539 m above mean sea level (AMSL)), situated between the Danube and Sava rivers, and the Vršac Mountains (641 m AMSL), located in the southeastern area of the region. Other significant geomorphological features include two sandy terrains: Banatska Peščara (Banat Sands, 250 m AMSL) in the southeast, and Bačka Peščara (Bačka Sands, 134 m AMSL) in the north, as well as lower areas and alluvial plains. Forests only comprise about 7% of Vojvodina and exist mainly in the mountains and terrains situated along river banks, while agricultural land occupies about 84% of the territory [40].

The climate of Vojvodina is moderate continental, with cold winters, and hot and humid summers. A wide range of extreme temperatures and a very irregular distribution of precipitation per month led to different values of calculated aridity types [17]. The climate diversity of Vojvodina is mainly the

result of surface wind blowing from opposite directions: NW when it is cold and humid, and SE when it is warm and dry [40,41]. The mean annual surface air temperature and the mean annual amount of precipitation are approximately 11.3 °C and 607.3 mm (see Section 4.1.5); [42–44], respectively.

Southeast Banat is the eastern subregion of Vojvodina, and possesses some distinctive geophysical characteristics. This area includes the highest altitude (Vršac Mountain) in Vojvodina as well as the lowest altitude (67–68 m AMSL) in the CB, which was found in the alluvial plain of the Nera and Karaš rivers. Furthermore, almost 20–30% of the area of Banatska Peščara is represented in this territory. Further south, SE Banat extends to the Danube, where the river is widest (3–7 km), while to the east, this area extends to the Nera river. The Danube and Nera rivers are the only mountain rivers located in Vojvodina, and they also form a natural, national border with Romania. It should be noted that the hydro-system Danube–Tisa–Danube [45] contributes to the greater hydrological diversity of SE Banat. Forests comprise about 20% [46] of SE Banat and mostly occur on Vršac Mountain, the sandy terrain of Banatska Peščara and on the alluvial plains of the rivers.

The climate of SE Banat is very much comparable to the climate of several other parts of the Vojvodina region, but with some distinctive differences. The values of the mean annual surface air temperature (11.7 °C), and the mean annual precipitation (652.4 mm) are higher than in Vojvodina as a whole (see Section 4.1.5), while the mean annual aridity is almost the same (see Section 4.1.1) in both the SE Banat subregion and Vojvodina as a whole. In SE Banat, there is also a higher frequency of the unique, dry, southeastern wind phenomenon known locally as a Košava, which occurs, on average, 129 days annually, while, in the rest of Vojvodina, it occurs at a lower rate of about 105 days annually [40].

*2.2. Data*

In this study, data from 15 meteorological stations were used (Figure 1), for the period 1949–2017. The following ten stations: Bački Petrovac, Bela Crkva, Kikinda, Palić, Rimski Šančevi, Senta, Sombor, Sremska Mitrovica, Vršac, and Zrenjanin were used for the territory of Vojvodina, while, for the subregion of SE Banat, the stations of Vršac and Bela Crkva, as well as the stations of Grebenac, Jasenovo, Kuštilj, Straža, and Vatin, were used.These stations are databases operated by the Republic Hydrometeorological Service of Serbia [47]. Data sets from each station were processed and analyzed in order to obtain the mean monthly values of surface air temperatures ($Tm$) and the monthly amount of precipitation ($Pm$). Correspondingly, the time series of $Tm$ and $Pm$ were used to calculate new time series for the aridity indices, and to analyze the trends in all time series. Prior to this analysis, the data homogeneity of the meteorological stations was examined using the Alexandersson test [48]. This test relies on the assumption that the difference/ratio between temperature/precipitation amounts at the station being tested and the reference series remains relatively constant over time. The correlation coefficients between the candidate stations and the reference stations were above 0.7, due to the relatively low and uniform flat terrain of the area being studied. The homogeneity analysis showed that the time series of the data for all the stations were homogeneous.

We also used the PDSI data network, developed by Dai et al. [49]. The PDSI network is basedon meteorological records of global land areas on a 2.5° × 2.5° grid points available at the National Center for Atmospheric Research [50]. The gridded PDSI values were extracted for the entire period 1927–2016 from a nearby grid point, which is closest to our tree ring sampling sites (Vlajkovac: 21.20° E, 45.07° N, and Šušara: 21.14° E, 44.94° N, Figure 1). It should be noted that the selected PDSI point can be considered representative for both regions, with SE Banat as the main area of investigation and Vojvodina as the control area.

In SE Banat, tree ring samples were collected (14–18 August 2018) in the villages of Vlajkovac and Šušara and their surroundings (Figure 1). We extracted 23 increment cores from 12 oak trees using an increment borer at chest height. At the sampling sites, the sandy and chernozem soil had good soil water conditions and abundant organic matter, resulting in a high canopy density.

## 3. Methods

### 3.1. The De Martonne Aridity Index

It has been almost a century since the French climatologist De Martonne created the DM Aridity Index. It has been used in many countries, including Greece [12], Turkey [9], Romania [13], Iran [51], Spain [14], as well as Serbia [17–19].

The annual and monthly values of the DM Aridity Index, $Ia_{DM}$ and $Im_{DM}$, can be represented by the following equations, respectively:

$$Ia_{DM} = \frac{Pa}{Ta + C},\tag{1}$$

$$Im_{DM} = \frac{Pm}{Tm + C},\tag{2}$$

where *Pa* and *Pm* are the annual and monthly amounts of precipitation, *Ta* and *Tm* are the mean annual and monthly surface air temperatures, and $C = 10\ °C$ is De Martonne's constant.

The classification of the De Martonne aridity climate is given in Table 1, with a total of seven types of aridity classes. As is evident, the humidity rises with an increase in the values of $Ia_{DM}$ and $Im_{DM}$, and vice versa. Using Equations (1) and (2), we supplemented the database with the time series of $Ia_{DM}$ and $Im_{DM}$ for each meteorological station.

**Table 1.** The De Martonne aridity index classification.

| Types of Climate | Values of $Ia_{DM}$ |
|---|---|
| Arid | $Ia_{DM} < 10$ |
| Semi-arid | $10 \leq Ia_{DM} < 20$ |
| Mediterranean | $20 \leq Ia_{DM} < 24$ |
| Semi-humid | $24 \leq Ia_{DM} < 28$ |
| Humid | $28 \leq Ia_{DM} < 35$ |
| Very humid | $35 \leq Ia_{DM} \leq 55$ |
| Extremely humid | $Ia_{DM} > 55$ |

According to the Dictionary of Statistical Terms [52], an index is a number formed from the ratio of aggregate values in the given period, to the aggregate values in the base period. If this definition is applied to $Ia_{DM}$ and $Im_{DM}$, according to Equations (1) and (2), we see that the $Ia_{DM}$ and $Im_{DM}$ represent numbers that provide the ratio of precipitation and temperatures in an annual or monthly period. Therefore, $Ia_{DM}$ and $Im_{DM}$ have dimensions of mm/°C. Hence, it is a physical value that is not a non-dimensional number, as it has been implicitly suggested in most scientific papers related to the DM Aridity Index. In this study, the numerical values of $Ia_{DM}$ and $Im_{DM}$ will be presented in the usual manner without the mentioning of dimensions.

### 3.2. Forestry Aridity Index

The annual value of the Forestry Aridity Index, *FAI*, by Führer et al. [31] is defined as

$$FAI = C_g \times \frac{T_{VII-VIII}}{(P_{V-VII} + P_{VII-VIII})},\tag{3}$$

where $T_{VII-VIII}$ is the average temperature in July and August in °C, $P_{V-VII}$ is the precipitation total in the period from May to July, and $P_{VII-VIII}$ is the precipitation total for the period of July–August, both in mm, and $C_g = 100\ mm/°C$ is the constant. By introducing the constant $C_g$ with dimensions [33], *FAI* lost its dimensions and became a "true" index, i.e., a non-dimensional number. With this modification in

the dimensions of the constant, $C_g$, the *FAI* has a better physical basis, and the interpretation of the results will not change.

The *FAI* and the average weather conditions of four different climate categories, as applied in forestry practice, are shown in Table 2. Unlike the De Martonne Aridity Index, in this case, humidity rises with a decrease in the value of *FAI* and vice versa. Using Equation (3), we supplemented the database with the time series of *FAI* for each meteorological station.

**Table 2.** Meteorological features of forestry climate categories [31].

| *FAI* Values | Forestry Climate Categories |
|:---:|:---:|
| Less than 4.75 | Beech climate |
| 4.75–6.00 | Hornbeam-oak climate |
| 6.00–7.25 | The sessile oak/Turkey oak climate |
| More than 7.25 | Forest-steppe climate |

In [32], it is shown that the lowest *FAI* values (high humidity) are spread throughout SE Banat, where the largest forest area in the Vojvodina region is found. It seems that the *FAI* can be a very good tool for presenting climate conditions during annual forest growth, something particularly important for those involved in forestry and agriculture.

### 3.3. The Palmer Drought Severity Index

The Palmer Drought Severity Index, PDSI, is a drought index based on a soil water balance equation [22], which measures the balance between moisture demand (evapotranspiration being driven by temperature) and moisture supply (precipitation). Values being taken into account in the calculation of the PDSI include monthly precipitation, potential, and actual evapotranspiration, infiltration of water into a given soil zone and runoff. The related equation is as follows:

$$X_t = p \times X_{t-1} + q \times Z_t, \tag{4}$$

where $X_t$ and $X_{t-1}$ are PDSI values for the actual and previous month, respectively. The $p$ (0.897) and $q$ (1/3) are coefficients or duration factors, demonstrating how sensitive the PDSI is to the monthly moisture anomaly $Z_t$ and how much autocorrelation the PDSI has. The detailed calculation of PDSI can be found in Palmer [22] and Dai [53].

### 3.4. The Tree Ring Width Index

After air-drying and sanding, the tree ring samples were processed following standard dendrochronological practices [54]. After rigorous cross-dating, the tree ring widths were measured under a binocular stereoscope using a LINTAB 6 measuring table (Rinntech, Heidelberg, Germany; the precision of 0.01 mm), after which we checked the quality of the cross-dating using the COFECHA method [55]. The measurements taken showed that there were no missing rings in the tree ring samples. The tree ring width measurements were standardized by detrending, in order to remove the effects of biological growth trends (e.g., the juvenile effect) as well as other low-frequency variations that result from stand dynamics. All series were conservatively detrended using a negative exponential function or a linear regression function with a negative slope. All detrended series were averaged to produce the final chronology by calculating bi-weighted, robust means to reduce the influence of outliers [56]. Variance stabilization was applied to minimize the effect of differences in the sample size over time, following the methods of Osborn et al. [57]. The standardization of ring-width data and the construction of the chronology were performed using ARSTAN (AutoRegressive STANdardization) method [56]. The reliability of the chronology was evaluated via the inter-series correlation coefficient (Rbar) and the expressed population signal (EPS) [58]. Both Rbar and EPS were computed for a 30-year

period, with a 15-year lag. In order to determine the most reliable period within the chronology, we used a threshold of EPS > 0.85 [58]. To retain more of the low-frequency climatic information, we used the standard chronology to detect the tree growth responses to climate parameters.

The statistical characteristics of our standardized chronology are summarized in Table 3. Beginning in 1920, the EPS value for ring width was greater than 0.85. The mean inter-series correlation coefficient was 0.40, indicating a sensitive measurement series where the tree ring samples contained a high degree of common climatic signals [59], and can therefore provide useful information on climate change in the location being studied.

**Table 3.** Statistical characteristics of the standardized tree ring chronology.

| Site | SE Banat |
| --- | --- |
| Altitude (m AMSL) | 83–181 |
| Standard deviation (SD) | 0.29 |
| First-order autocorrelation (AC1) | 0.18 |
| Length of chronology (CL, years) | 99 |
| Mean tree ring width (mm) | 3.31 |
| Mean inter-series correlation (Rbar) | 0.40 |
| Expressed population signal (EPS) | 0.92 |

*3.5. Data Analysis*

For data analysis, (a) the linear trend, as an indicator of the recent change in values, was used, (b) as an indicator of the link of different values, the correlation coefficient was used, and (c) in some cases, to determine their similarity, time series were directly graphically compared.

For analyzing the trends of the $Ia_{DM}$, *FAI*, PDSI, and TRWI, as well as the mean annual surface air temperatures, *Ta*, and the annual amount of precipitation, *Pa*, three statistical approaches were used. These are: (*i*) *the tendency (trend) equation*, (*ii*) *the Mann–Kendall (MK) test*, and (*iii*) *the trend magnitude*.

*The tendency (trend) equation* was calculated and plotted for each time series using linear interpolation [60]. This approach yields results, on the basis of the slope and the coefficient direction of the trend equation, which are simple to interpret, both analytically and graphically. In this kind of interpretation, where the slope (and the coefficient direction of the trend equation) is greater than zero, less than zero or equal to zero, the sign of the trend is *positive* (increasing), *negative* (decreasing) or *there is no trend* (no change), respectively.

*The MK test* is used [61–64] for estimating all trends. According to the MK test, two hypotheses were tested: the null hypothesis, H0, that *there is no trend* in the time series; and the alternative hypothesis, Ha, that *there is a significant trend* in the time series, for a given $\alpha$ significance level [65]. Probability, *p*, was calculated in percentages [42,43,66,67] in order to determine the level of confidence in the hypothesis. If the computed value *p* is lower than the chosen significance level $\alpha$ (e.g., $\alpha$ = 5%), then H0 (*there is no trend*) should be rejected, and the Ha (*there is a significant trend*) should be accepted. In cases where *p* is greater than the significance level $\alpha$, H0 (*there is no trend*) cannot be rejected.

MK tests are used widely in the environmental sciences, assisting in measurements and analyses of temperature, precipitation, sunshine hours, cloud cover, relative humidity and wind speed [68]; temperature and precipitation [66]; precipitation [69]; extreme temperatures [42,43,67,70]; hail [71–74]; aridity [17–19,32,75]; evapotranspiration [76]; and atmospheric deposition [77]. MK tests are simple, robust, and can cope with missing values. The MK test is a non-parametric test that does not provide explicit information on the statistical significance of the parametric approach of linear regression trends.

*Trend magnitude* is defined as the difference in values between the beginning and ending of a period for which a trend equation is being calculated, and that has been obtained through the use of a linear trend equation [42,43]. With regards to the trend magnitude, we noted the following: the trend magnitude increases with the increase of this difference in values, and, conversely, decreases with the decrease of this difference in values. Likewise, the sign of the difference can determine the trend sign,

for when the difference is greater than zero, less than zero, or equal to zero, the sign of the trend is negative (decrease), positive (increase), or there is no trend (no change).

Here, the magnitude trend will not be used for direct trend estimation but will instead be used to estimate the influence of the trend magnitudes of temperature and precipitation on the aridity trend.

### 3.6. Software

The mean surface air temperatures, amount of precipitation, DM aridity indices, trend equations, and linear trend lines were computed and plotted for each time series, using the program Microsoft Excel (v. 2010, Seattle, WA, USA). Results were interpolated and mapped by Kriging method, using the ArcMap. For calculating the probability, $p$, and hypothesis testing, XLSTAT's statistical analysis software was used [78].

## 4. Results

### 4.1. Annual Distributions and Trends

#### 4.1.1. The De Martonne Aridity Index

The distribution of the mean annual De Martonne Aridity Index, $Ia_{DM}$, and its trend for the Vojvodina region and SE Banat region, during the period 1949–2017, are presented in Figure 2.

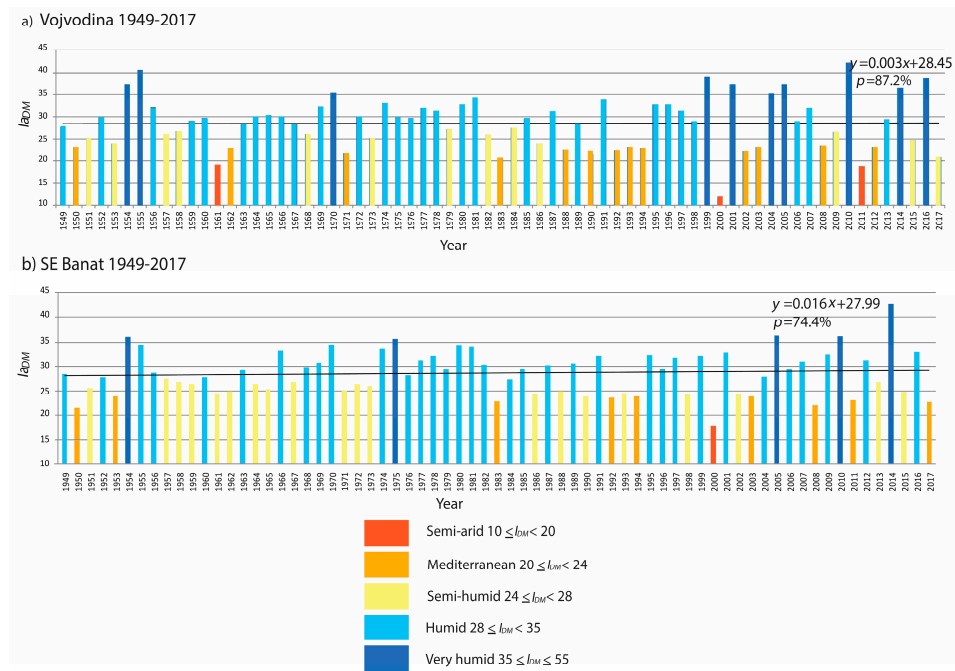

**Figure 2.** The distribution and trend of the mean annual De Martonne Aridity Index, $Ia_{DM}$, for (**a**) Vojvodina and (**b**) SE Banat both in the period 1949–2017. The aridity trend is presented with a solid, black line and trend equation, and $p$ is the probability.

The graphs and equations in Figure 2a,b shows that the $Ia_{DM}$ trends are positive. As both values of $p$ (87.2% and 74.4%) are greater than $\alpha$ (5%), H0 (*there is no trend*) cannot be rejected according to the MK test. The risks of rejecting H0 while it is true are 87.2% and 74.4%. The presented values, when referencing the mean annual De Martonne Aridity Index, $Ia_{DM}$, suggest that there are no changes in aridity in Vojvodina and SE Banat in the investigated period 1949–2017.

It should be noted that the minimum values of the $Ia_{DM}$ for both the Vojvodina (12.70) territory and SE Banat (17.73) subregion were reached in 2000, while the maximum values for the Vojvodina territory (42.22) and SE Banat (42.68) subregion were reached in 2010 and 2014, respectively. It appears

that, at the beginning of the 21st century, there existed a greater variation in aridity than in the previous century. All maximum and minimum values were recorded in this century, and so it appears that a semi-arid climate existed only in Vojvodina during the last century (1961).

The mean values of the $Ia_{DM}$ for Vojvodina and SE Banat are 28.55 and 28.56, respectively. Therefore, in both territories, the mean type of aridity climate is humid, but quite close to semi-humid, as the value of 28 is used to distinguish these two types.

### 4.1.2. The Forestry Aridity Index

The distribution of the mean annual Forestry Aridity Index, *FAI*, and its trend for Vojvodina and SE Banat, for the period 1949–2017, are presented in Figure 3.

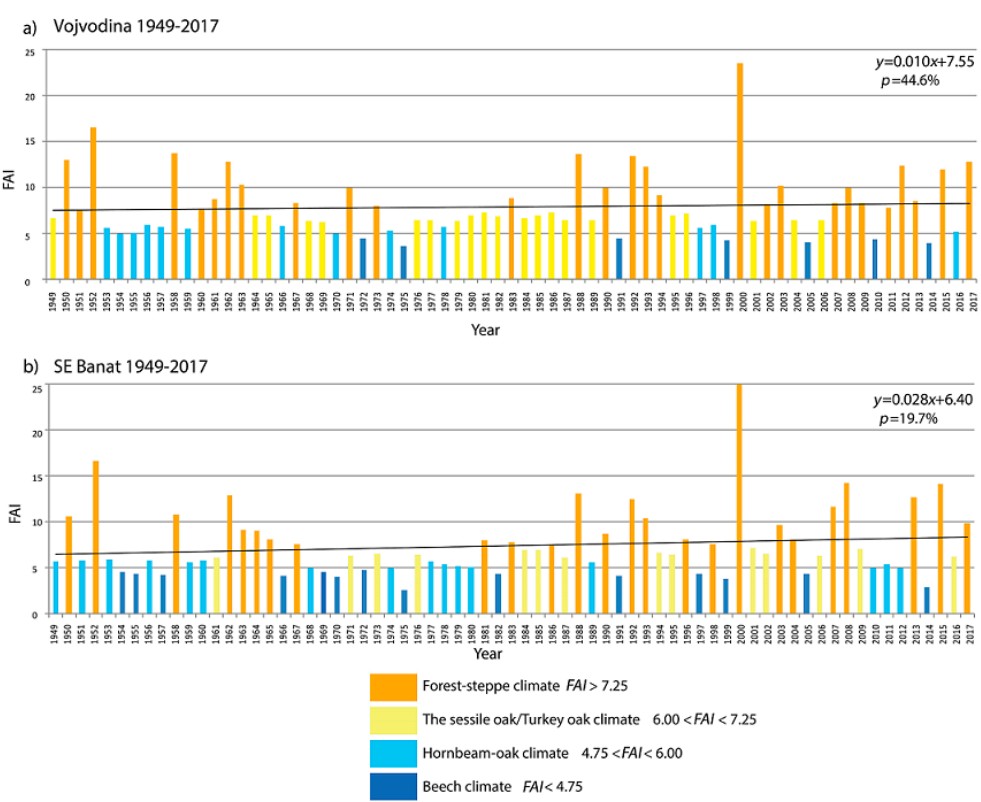

**Figure 3.** The distribution and trend of the mean annual Forestry Aridity Index, *FAI*, for (**a**) Vojvodina and (**b**) SE Banat both for the period 1949–2017. The aridity trend is presented with a solid, black line and trend equation, and *p* is the probability.

Yet again, we can see that both *FAI* trends are positive. As both values of *p* (44.6% for Vojvodina and 19.7% for SE Banat) are greater than $\alpha$ (5%), H0 (*there is no trend*) cannot be rejected. The presented values, when referencing the mean annual Forestry Aridity Index, *FAI*, suggests that there are no aridity changes in Vojvodina and SE Banat in the investigated period 1949–2017.

It should be noted that extreme values of the *FAI* for both the Vojvodina territory and the SE Banat subregion (maximums being 23.30 and 24.90, and minimums being 3.73 and 2.45) were reached in the same years, 2000 and 1975, respectively.

### 4.1.3. The Palmer Drought Severity Index

The distribution and trend of the Palmer Drought Severity Index, PDSI, for the lat–long point (21° E, 45° N), representing the regions of SE Banat and Vojvodina in the period 1927–2016 are presented in Figure 4.

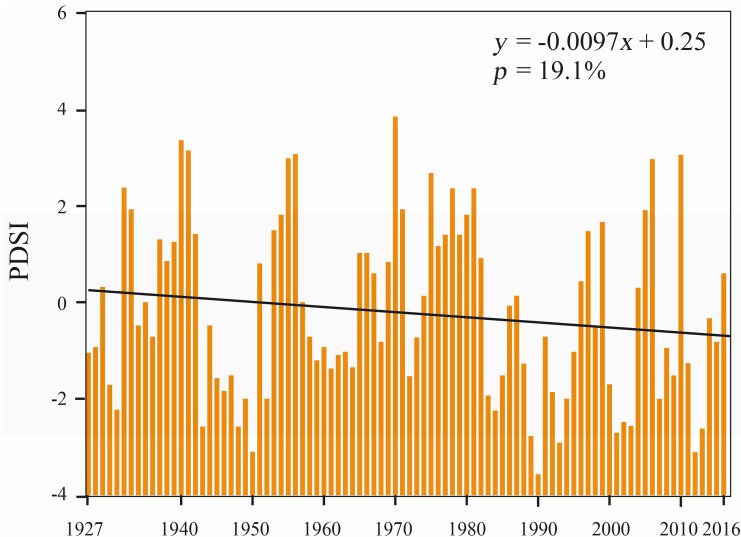

**Figure 4.** The distribution and trend of the PDSI for SE Banat and Vojvodina in the period 1927–2016. The PDSI trend is presented with a solid, black line and trend equation, and *p* is the probability.

As indicated, the PDSI trend is negative. Since the computed probability value *p* of PDSI (19.1%) is higher than the significance level, $\alpha$ (5%), H0 (*there is no trend*) should be accepted. It indicates that there is no significant PDSI trend for SE Banat and Vojvodina.

The average value of PDSI is approximately 0.0. Around this value, the annual values within the extremes that were recorded in 1970 (PDSI$_{max}$~4.00) and 1990 (PDSI$_{min}$~−3.75), seemingly vary chaotically.

### 4.1.4. The Tree Ring Width Index

The annual distribution and trend of the Tree Ring Width Index, TRWI, for SE Banat, in the period 1927–2017, are presented in Figure 5.

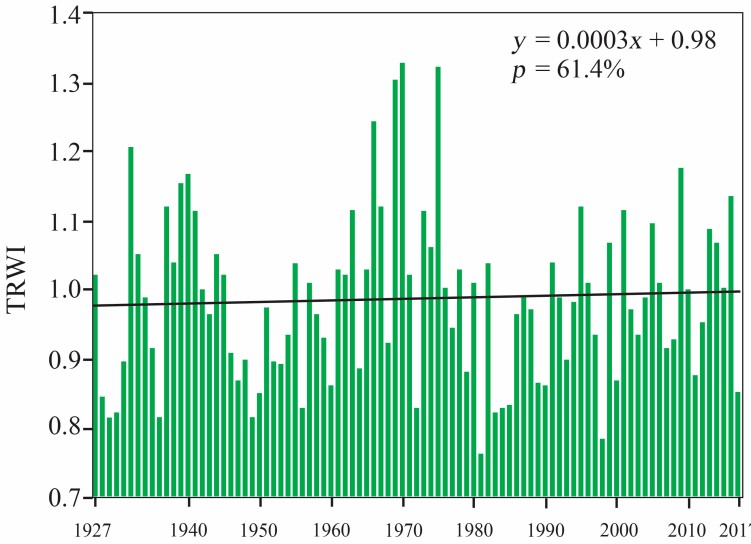

**Figure 5.** The distribution and trend of the TRWI for SE Banat in the period 1927–2017. The TRWI trend is presented with a solid, black line and trend equation, and *p* is the probability.

The results suggest that the TRWI trend is positive. The computed probability values *p* of TRWI is 61.4%, which is higher than the significance level, $\alpha$ (5%). Therefore, H0 (*there is no trend*) should be accepted. The values indicate that there is no significant TRWI trend for SE Banat.

The annual values of TRWI oscillate around the mean value (~1.00) ranging from ~0.75 to ~1.34, while marked increases in TRWI values are observed in the 1960s and 1970s.

### 4.1.5. Temperature and Precipitation

The distribution of the mean annual temperature, *Ta*, and the mean annual amount of precipitation, *Pa*, for Vojvodina and the SE Banat, in the period 1949–2017, are presented in Figure 6.

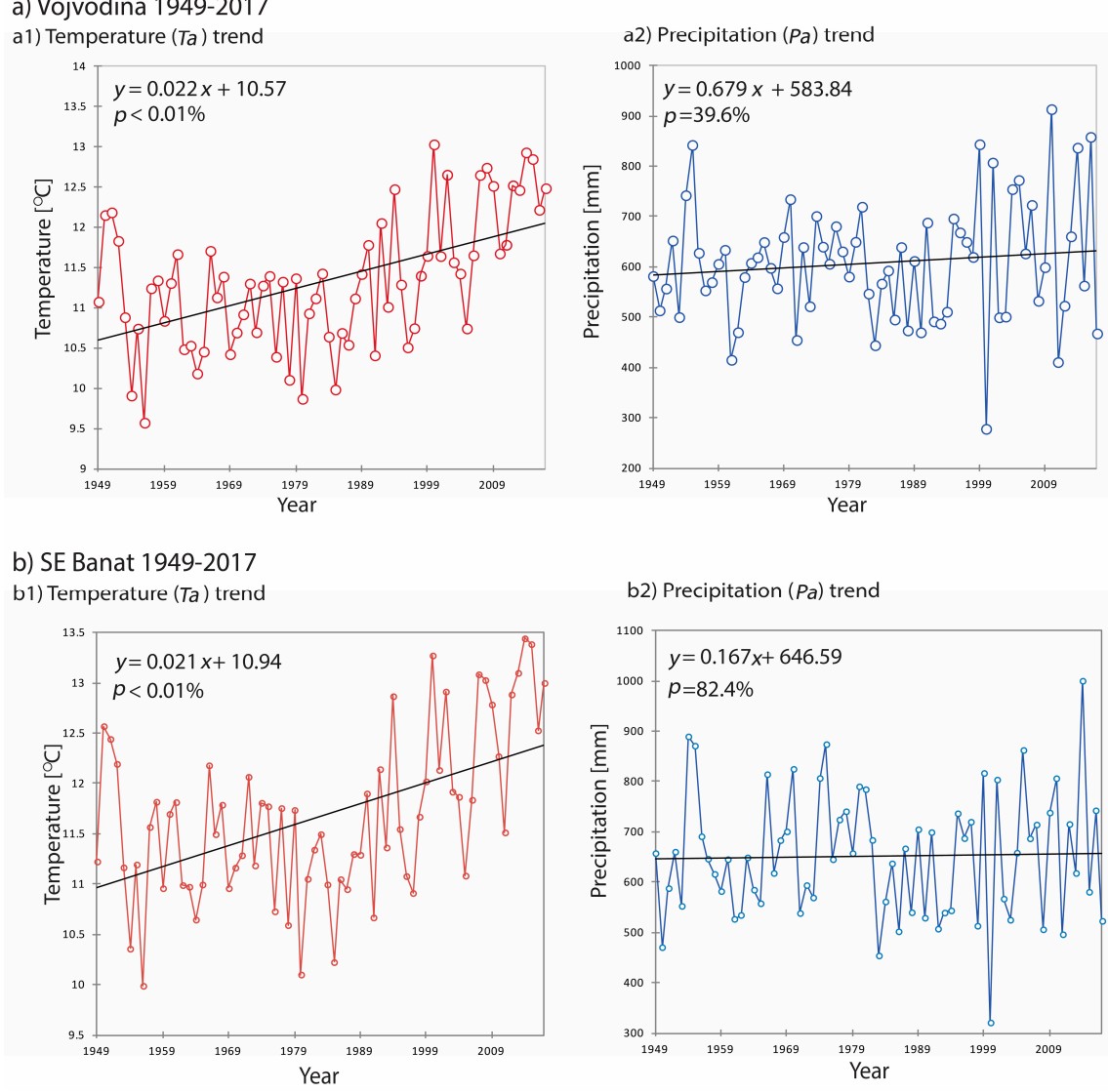

**Figure 6.** The distributions and trends of the mean annual temperature, *Ta*, and the mean annual amount precipitation, *Pa*, in the period 1949–2017, for (**a**) Vojvodina and (**b**) SE Banat, respectively. Trends of the *Ta* and the *Pa* are presented with solid, black lines and trend equations, and *p* are probabilities.

Figure 6(a1,b1) shows that *Ta* trends are positive. Since the computed probability values *p* of the *Ta* for Vojvodina (0.01%) and SE Banat (0.01%) are lower than the significance level, *α* (5%), Ha (*there is a significant trend*) should be accepted. In accordance with the MK tests, *Ta* trends in Vojvodina and SE Banat are declared as significant positive trends.

Figure 6(a2,b2) shows that *Pa* trends are positive. As values *p* of *Pa* for Vojvodina (39.6%) and SE Banat (82.4%) are higher than *α* (5%), H0 (*there is no trend*) cannot be rejected. The results of the MK tests, when considering the mean annual amount of precipitation, *Pa*, suggest that there are no trends in precipitation in Vojvodina and SE Banat, in the period 1949–2017.

It should be noted that lowest values of the *Ta* for Vojvodina (9.6 °C) and SE Banat (9.9 °C) territories were reached in 1956, while the highest values of the *Ta* for Vojvodina (13.0 °C) and SE Banat (13.4) were reached in 2010 and 2014, respectively. The mean values of the *Ta* for Vojvodina and SE Banat are 11.3 °C and 11.7 °C during the period 1949–2017, respectively.

However, in both Vojvodina and SE Banat, in the instrumental period 1949–2017, significant positive temperature trends and an increase in temperatures by 1.5 °C and 1.4 °C, respectively, were calculated. These values were obtained on the basis of the definition of the trend magnitude.

The minimum and maximum of *Pa*, for the territory of Vojvodina, occurred in the years 2000 (278.1 mm) and 2010 (913.7 mm), respectively, while in SE Banat extreme values of *Pa* were reached in the years 2000 (321.0 mm) and 2014 (100.5 mm). The mean values of *Pa* for Vojvodina and SE Banat are 607.3 mm and 652.4 mm.

## 4.2. Correlation Analysis

In order to evaluate the response of tree ring width to the measured temperature and precipitation quantities, as well as calculated climate variables (aridity) from SE Banat, correlation coefficients were calculated. The results indicate that the dominant factor in annual radial growth was increased seasonal precipitation, which is in accordance with the statistically significant positive correlation between tree ring width and precipitation (Figure 7). The mean precipitation from April to August had the strongest influence on radial growth ($r = 0.53$, $p < 0.0001$), but there was no significant correlation with temperature (Figure 7).

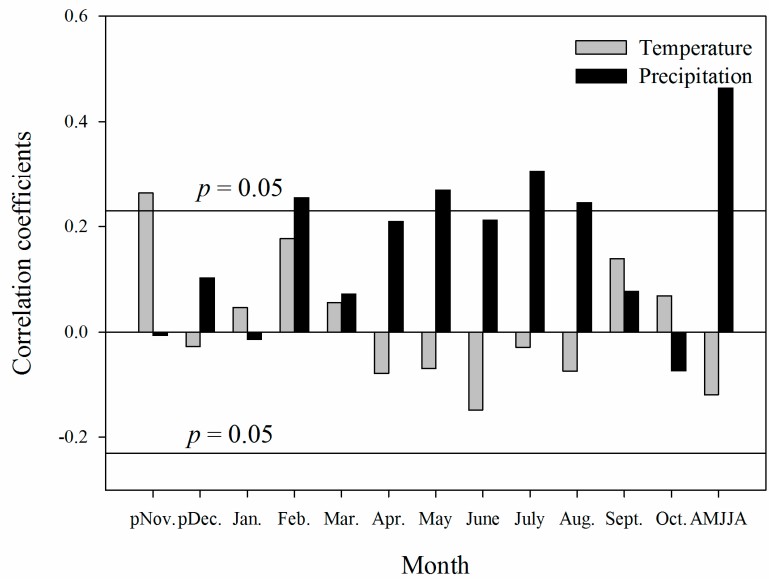

**Figure 7.** The correlation coefficients between the TRWI and the mean monthly temperature, *Tm*, and the mean monthly amount of precipitation, *Pm*, for SE Banat during the period 1949–2017, where *p* is the significance level, the prefix "p" indicates the previous year and "AMJJA" represents the period from April to August in the current year.

The TRWI shows a satisfactory correlation between the PDSI ($r = 0.51$, $p < 0.001$, Figure 8) and the $Im_{DM}$ ($r = 0.46$, $p < 0.001$, Figure 9), from April to August, suggesting that the linear dependence reflected by the index is relatively high, despite the index values being the result of highly nonlinear processes, while the TRWI shows negative and weak correlation with *FAI* ($r = -0.30$, $p < 0.010$) (Figure 10).

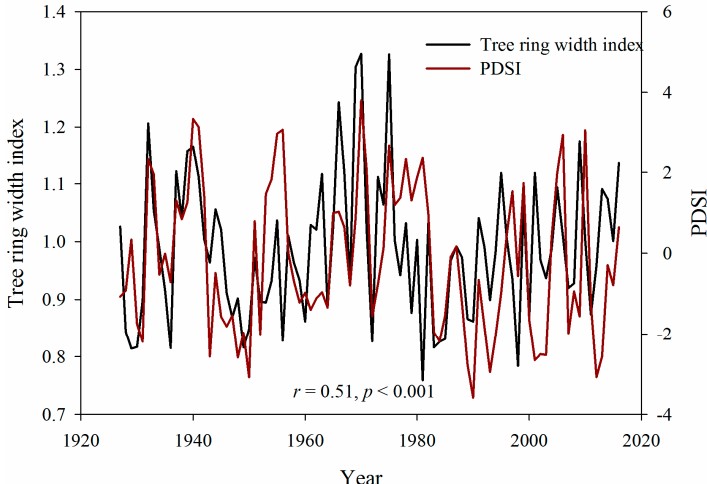

**Figure 8.** Comparisons between the TRWI and the PDSI during April–August for SE Banat from 1927 to 2016, where *r* is the coefficient of correlation and *p* is the significance level.

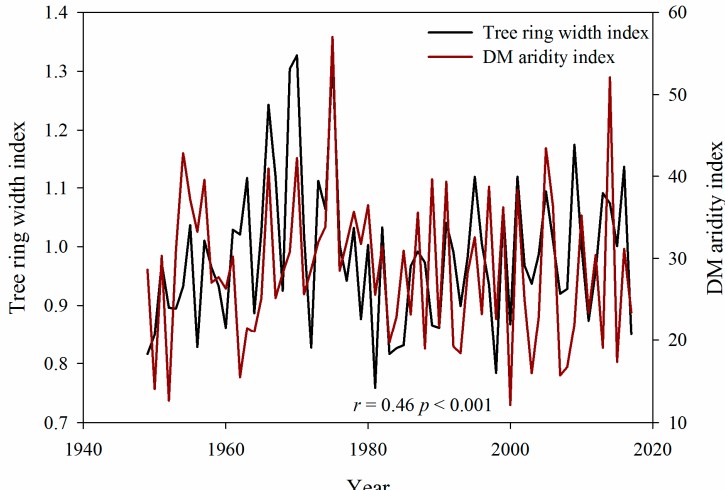

**Figure 9.** Comparisons between the Tree Ring Width Index and the DM Aridity Index during April–August for SE Banat from 1949 to 2017, where *r* is the coefficient of correlation and *p* is the significance level.

However, the TRWI shows a significant correlation with the $Im_{DM}$ and the PDSI, during the April–August period (Figures 8 and 9). The *FAI* highlights the temperature [31,32], while the $Im_{DM}$ highlights the precipitation [17]. When considering the significant negative correlations between the TRWI and the precipitation, it follows that the response of the TRWI to the $Im_{DM}$ is higher than to the *FAI*.

A visual comparison of the TRWI values with the PDSI (Figure 8) and the DM Aridity Index (Figure 9), both during the April–August period, and the *FAI* (Figure 10), shows significant similarities. Extreme "spikes" and changes between them overlap somewhere.

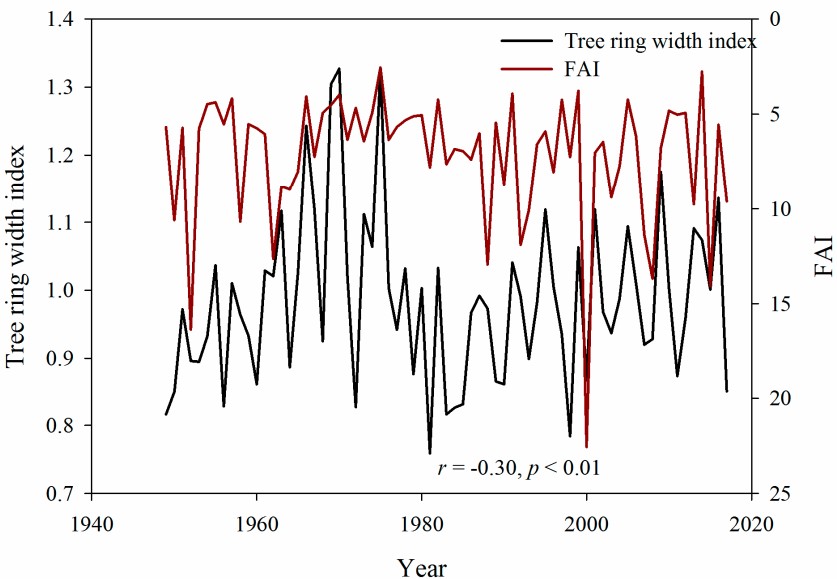

**Figure 10.** Comparisons between the TRWI and the *FAI* for SE Banat from 1949 to 2017, where *r* is the coefficient of correlation and *p* is the significance level. The vertical axis of the *FAI* has been reversed to compare.

## 5. Discussion

In this study, tree ring, meteorological, and aridity data were investigated independently in Southeast Banat, during the period 1927–2017. Tree ring samples were extracted (23 increment cores) from 12 oak trees in two locations in SE Banat. From these samples, the Tree Ring Width Index, or TRWI, was calculated and analyzed. The mean annual surface air temperature, *Ta*, and the mean annual amount of precipitation, *Pa*, were analyzed from seven meteorological stations in SE Banat, for the period 1949–2017. Using the same meteorological data, the DM aridity index and the Forestry Aridity Index, or *FAI*, were derived and analyzed. In addition to the above-mentioned indices, the Palmer Drought Severity Index, or PDSI, is assumed as an independent index from a grid point that belongs to the SE Banat territory. This data was analyzed for the period 1927–2016.

As a control territory for benchmarking the climate parameters of SE Banat, the larger territory of Vojvodina was chosen. Both territories have almost identical geomorphological and climatological characteristics. In the case of Vojvodina, only the TRWI parameter was not available. The DM Aridity Index, *FAI*, *Ta*, and, *Pa* were all counted as original data based on the meteorological data from 10 meteorological stations, in the period 1949–2017. The PDSI had the same value as it did in SE Banat.

Table 4 shows that, of the six analyzed climatic parameters for SE Banat, only the temperature had a positive trend in the period 1949–2017, while the other five parameters displayed no trend. Moreover, the TRWI and the PDSI displayed no trend for the periods 1927–2017 and 1927–2016, respectively, which is almost 20 years longer than the period during which the meteorological data were being recorded. The control territory of Vojvodina shows the same results. The five climatic parameters showed no trend in four cases, and only in the case of temperature could a trend be determined.

As seen in Table 4, in both areas, significant trends in temperature, as well as an increase in temperature reaching 11.3 °C and 11.7 °C were recorded. Trends in precipitation were not recorded, and there were no trends in the DM Aridity Index, the *FAI*, or the PDSI either.

Taking trend equations into account, with their coefficients of linear regression, as well as taking into account the MK test results, it can be concluded that trends of the $Ia_{DM}$ and the *FAI* are negligibly small, recording values of nearly zero and showcasing that there are basically no trends. Considering the $Ia_{DM}$ and the *FAI* in this manner, it seems that there is no change in aridity in Vojvodina for the period 1949–2017. In Hrnjak et al. [17], very similar results in aridity trends with the De Martonne's aridity index and the Pinna combinative index were obtained, for the territory of Vojvodina, for the

period 1949–2006. Very similar results in aridity trends for the previous two indices have been obtained for other Serbian regions, namely Kosovo and Metohija [18] and Central Serbia [19], where it has also been found that there is no trend. The same result, but without the *FAI* trend, in Vojvodina was obtained for the period 1949–2006 [32].

**Table 4.** Trends of climate parameters for SE Banat and Vojvodina.

| Ordinar | Climate Parameters | Periods | Trends | |
|---|---|---|---|---|
| | | | SE Banat | Vojvodina |
| 1. | TRWI | 1927–2017 | No | - |
| 2. | *Ta* | 1949–2017 | Yes | Yes |
| 3. | *Pa* | 1949–2017 | No | No |
| 4. | DM Aridity Index | 1949–2017 | No | No |
| 5. | *FAI* | 1949–2017 | No | No |
| 6. | PDSI | 1927–2016 | No | No |

## 6. Conclusions

It appears that the change in aridity/drought trends cannot be the result of only a change in temperature trends. The change in temperature trends is not enough, in and of itself, to trigger a change in the aridity trend. It appears that some critical values in the temperature trend and/or the precipitation trend are existing, potentially leading to a change in the aridity trend. At this point, we were not able to identify these values with absolute certainty, and so it remains for future research. Unlike the temperature, which shows variability in trends, the aridity did not show any change in trends.

The previous statement affirms that the TRWI's behavior is a useful independent indicator of climate change in the SE Banat region. This climatic parameter indicates a slight dependence on temperature. The TRWI does not exhibit any sensitivity to temperature change, showing poor correlation with it, while at the same time displaying significant correlation with precipitation. This suggests that a change in rainfall trend might cause changes in the aridity trend, another thing that should be more closely investigated in the future. Forest vegetation seems to be more susceptible to changes in rainfall trends, rather than changes in temperature trends. This is a testament to the significance of existing vegetation playing a major role in reducing climate extremes and ensuring that the climate is less susceptible to fluctuation.

Conclusions comparable to the ones reached in this study can be found in various paleoclimate studies [36], suggesting that aridity in the investigated region is not characteristic only for the recent climate, but was also common in past geological periods. Many analyzed paleoclimatic proxies related to the last glacial loess also report a periodic [36,79,80] or even continuous presence of arid climate conditions [81–83].

**Author Contributions:** M.B.G., Q.H., S.B.M., and Z.G. conceived and designed the study; W.A., C.X., M.B.G., S.B.M., and F.Y. performed the tree ring data sampling; W.A., C.X., F.Y., and Q.H. performed laboratory measurements and analyzes; M.B.G., W.A., C.X., M.G.R., and G.G. processed and graphically presented the data; M.B.G., W.A., M.G.R., G.G., and Z.P. analyzed the data; M.B.G., Z.P., W.A., C.X., and S.B.M. wrote the paper.

**Funding:** This study is funded by the First Program of the Chinese-Serbian Developing Projects, the National Key R&D Program of China, the Project of the Serbian Ministry of Education, the Natural Science Foundation of China, and the Chinese Academy of Sciences.

**Acknowledgments:** This study is supported by the First Program of the Chinese-Serbian Developing Projects (Project title: "A Comparative Study of Past Climate Change in the East Asian Monsoon Region and the Westerly Zone Using Multiple Timescales"). Additionally, the investigations presented here were supported by the National Key R&D Program of China (Grant No. 2017YFE0112800), Project 176020 of the Serbian Ministry of Education, the Natural Science Foundation of China (41690114, 41888101) and the Chinese Academy of Sciences (CAS) Pioneer Hundred Talents Program. The authors are grateful for the support of Valentina Janc and Joži Dani. The authors appreciate the suggestions of the anonymous reviewers that led to improvements in the paper.

**Conflicts of Interest:** The authors declare no conflict of interest. The founding sponsors had no role in the design of the study; in the collection, analyses, or interpretation of data; in the writing of the manuscript, and in the decision to publish the results.

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
