# Peer review of "Independent Aridity and Drought Pieces of Evidence Based on Meteorological Data and Tree Ring Data in Southeast Banat, Vojvodina, Serbia"

_atmosphere, doi:10.3390/atmos10100586_

Round 1
Reviewer 1 Report
Just some minor comments:
1) Line 22: change "acted" to "used"
2) Line 29-30: Temperature trends could be significant while there is no trend in other variables. There is no necessity for the temperature to trigger a trend in other variables. Thus, the use of "too insufficient to" has no relevance. Or, should there be any sufficient T trend to trigger a trend in another other stationary varialbe?
3) Line 189: insert at the end of line "or monthly"
4) Line 344: add the lat-lon coordinates for the PDSI point and revise the sentence to like: "The distribution and trend of the PDSI for the lat-lon point (?,?), representing the regions of SE Banat and ..."
5) Line 363: Should it be TRWI instead of PDSI?
6) Line 410: the numbers are different from what are shown in the figure.
7) Line 426-427: the two are not shown in the same figure, thus it is hard to tell about the overlapping.
Author Response
Responses have been attached.

Reviewer 2 Report
Please find my comments in the attached pdf file.
Data, methods and presentation of results are sound and well written.
One suggestion for further comparison of indices included.
Minor improvement of discussion and conclusion is necessary.

Author Response
Responses have been attached.
